# Cell Senescence-Independent Changes of Human Skin Fibroblasts with Age

**DOI:** 10.3390/cells13080659

**Published:** 2024-04-09

**Authors:** Nicola Fullard, James Wordsworth, Ciaran Welsh, Victoria Maltman, Charlie Bascom, Ryan Tasseff, Robert Isfort, Lydia Costello, Rebekah-Louise Scanlan, Stefan Przyborski, Daryl Shanley

**Affiliations:** 1Department of Biosciences, Durham University, Durham DH1 3LE, UK; 2Biosciences Institute, Newcastle University, Newcastle upon Tyne NE1 7RU, UK; james.wordsworth2@ncl.ac.uk (J.W.); cwelsh274@gmail.com (C.W.);; 3Proctor & Gamble, Cincinnati, OH 45201, USAisfort.rj@pg.com (R.I.)

**Keywords:** skin, senescence, ageing, extracellular matrix, TGF, fibroblast, myofibroblast, collagen, fibrosis, metalloproteinase

## Abstract

Skin ageing is defined, in part, by collagen depletion and fragmentation that leads to a loss of mechanical tension. This is currently believed to reflect, in part, the accumulation of senescent cells. We compared the expression of genes and proteins for components of the extracellular matrix (ECM) as well as their regulators and found that in vitro senescent cells produced more matrix metalloproteinases (MMPs) than proliferating cells from adult and neonatal donors. This was consistent with previous reports of senescent cells contributing to increased matrix degradation with age; however, cells from adult donors proved significantly less capable of producing new collagen than neonatal or senescent cells, and they showed significantly lower myofibroblast activation as determined by the marker α-SMA. Functionally, adult cells also showed slower migration than neonatal cells. We concluded that the increased collagen degradation of aged fibroblasts might reflect senescence, the reduced collagen production likely reflects senescence-independent processes.

## 1. Introduction

Skin is the largest organ in the human body, performing numerous functions beyond that of a barrier to environmental stress. For example, skin is involved in sensory perception, thermoregulation, and immunosurveillance. Like other tissues, skin is subject to intrinsic ageing, its ability to perform its functions diminishing as a result of these intrinsic changes (as well as a strong extrinsic component) [1]. Phenotypically, old skin is dry, rough, and itchy, with uneven pigmentation, reduced capacity for wound healing, wrinkles, and impaired collagen and elastin networks [2]. Old skin has diminished hair growth and sebaceous gland function, flatter dermal papillae, reduced melanocyte concentrations and less cellular turnover compared with young skin [3,4,5]. Skin ageing is the most visible aspect of ageing and the resulting phenotypic changes impact both physiological and psychological well-being [6]. 

Skin is a multi-layered tissue composed of an outer epidermis and an underlying dermis. The epidermis is comprised mainly of keratinocytes which are avascular and form a highly organized and stratified structure, with proliferative basal keratinocytes differentiating as they progress towards the skin surface. Beneath the epidermis lies the dermal-epidermal junction which is a thin basement membrane that enables communication between the dermis and epidermis. The dermis is a vascular, cell-sparse tissue composed mostly of a fibrous connective tissue known as the dermal extracellular matrix (ECM). Dermal tissue is essential for providing structural integrity to the skin and nourishing support for the epidermis [7].

Dermal fibroblasts are responsible for synthesising ECM components as well as proteins that degrade them; the balance between these two opposing functions regulates the ECM in both healthy skin and damaged or aged skin. Fibroblasts respond to local mechanical forces [8,9] as well as biochemical cues such as stimulation by TGF-β [10], which at higher levels can induce differentiation into myofibroblasts [11]. Myofibroblasts are metabolically active cells that secrete high levels of ECM proteins such as collagens I and III [12]. The composition of the dermal microenvironment changes as we age. Fibroblasts in young tissue reside in an environment under mechanical tension which arises from physical binding to the ECM [13], but with age collagen fibrils become increasingly fragmented and fewer in number [14], resulting from both decreased production and enhanced degradation by matrix metalloproteinases (MMPs) such as MMP1 [8,13,15,16]. Currently, little is known about how the capacity of fibroblasts to respond to TGF-β changes with age, including myofibroblast differentiation; however, the role of inflammatory stimuli in myofibroblast activation is well evidenced [17]. Outside the inflammageing model, ageing may reduce hyaluronan synthesis, causing the loss of CD44/epidermal growth factor (EGF) receptor signalling, and resulting in impaired TGF-β1 dependent fibroblast activation [18].

The aged dermal environment also contains an increasing population of senescent fibroblasts [19,20,21], and it is becoming increasingly accepted that “skin aging is significantly enforced by the accumulation of senescent dermal fibroblasts” [22]. There have been various mechanisms and implications proposed for the effect of senescent cells in human skin [23] and for ageing in general [21,24,25,26], although the evidence is not conclusive [27]. Inflammatory signals are thought to activate myofibroblasts, which can be inhibited by activation of anti-ageing molecules such as AMPK [17,28]. 

Here, we compared ageing and senescent fibroblasts to assess their impact on TGF-β signalling and myofibroblast differentiation, and our results indicated that while senescent cells may contribute to increased matrix degradation, they did not affect the depleted response of adult fibroblasts to TGF-β or their reduced myofibroblast differentiation, which seemed instead to reflect an altered TGF-β homeostasis. 

## 2. Materials and Methods

### 2.1. Cell Culture

Cultured fibroblast lines included three independent human neonatal dermal fibroblasts (HDFn) labelled A (Caucasian male, catalogue number: C-004-5C, lot number: #1366434), B (Caucasian male, catalogue number: C-004-5C, lot number: #1366356), and C (Caucasian male, catalogue number: C-004-5C, lot number: #1206197), and three independent adult fibroblast lines labelled G (55 years old Caucasian female, catalogue number: C-013-5C, lot number: #1528526), H (60 year old, Caucasian male, catalogue number: A11634, lot number: #1090465), I (65 year old, Caucasian female, catalogue number: A11636, lot number: #200910-901) purchased from Life Technologies (Carlsbad, CA, USA), and one adult cell line lablled J (59 year old, Caucasian female, catalogue number: CC-2511, lot number: #693503) purchased from Lonza (Basel, Switzerland). 

Senescence was induced in cells with 20Gy X-irradiation ten days prior to seeding. At this dose, 100% of fibroblasts become senescent within 1–2 replications, as we have previously shown through the abolition of cell proliferation and labelling indices for BrdU and Ki67, as well as increased senescence-associated β-galactosidase (Sen-β-Gal) [29,30]. Cells were seeded into standard tissue culture-treated 12-well dishes at a density of 10,000 cells/cm^2^, proliferative, or 65,000 cells/cm^2^, senescent, in 3.5 mL M106 medium (ThermoFisher Scientific, catalogue number: M106500) supplemented with low serum growth supplement (LSGS, ThermoFisher Scientific, catalogue number: S00310) at 37 °C, 5% CO_2_ for 4 days.

### 2.2. Treatment Protocol

Cells from each cell line were serum starved 24 h prior to treatment by removing LSGS supplementation from the media. Following 24 h of incubation at 37 °C and 5% CO_2_, cells were assigned to one of three treatments: baseline, control, or TGF-β. Baseline samples were not treated in any way prior to harvesting at the experiment start (0 h). TGF-β and control samples were treated with media containing 5 ng mL^−1^ TGF-β1 reconstituted in 10 mM citric acid/0.1% BSA or 10 mM citric acid/0.1% BSA vehicle control, respectively. In control and TGF-β groups, cells were harvested at 0.5, 1, 2, 3, 4, 8, 12, 24, 48, 72, 96 h post-treatment. All 216 conditions were repeated six times resulting in 1296 individual samples. Samples were shipped to Procter and Gamble (P&G), Cincinnati for quantification on a high throughput PCR Smart chip platform by WaferGen.

### 2.3. Co-Culture

To produce conditioned media either proliferative or irradiation-induced senescent fibroblasts were cultured with 7 ml serum-free media for 3 days. As per normal treatment protocol, fibroblasts were serum starved for 24 h at 37 °C and 5% CO_2_, prior to treatment with TGF-β, by removing LSGS supplementation from media. Serum-free media was then replaced with a 50:50 mix of conditioned media and fresh serum-free M106 containing 5 ng mL^−1^ TGF-β1 reconstituted in 10 mM citric acid/0.1% BSA. Cells were harvested at 96 h post-treatment, replacing the conditioned media with fresh conditioned media at 48 h.

### 2.4. 3D Cell Culture and Treatment Protocol

Human male neonatal (<14 days), young (<30 years), middle-aged (40–45 years), and old (60+ years) dermal fibroblasts (Life Technologies) were maintained in Dermal Fibroblast Growth Medium (DFGM) comprised of Medium 106 (Thermo Fisher Scientific, Waltham, MA, USA), supplemented with LSGS (Thermo Fisher Scientific), 10 μg mL^−1^ gentamicin, and 0.25 μg mL^−1^ amphotericin B (Thermo Fisher Scientific) at 37 °C in a 5% CO_2_ humidified incubator following the supplier’s instructions.

Dermal models were generated by seeding HDFs (5 × 10^5^ cells) onto inert porous polystyrene membranes (12-well Alvetex^®^ scaffold inserts, Reprocell Europe Ltd., Glasgow, UK) and incubating at 37 °C in a 5% CO_2_ humidified incubator in DFGM supplemented with 5 ng mL^−1^ TGF-β1 (Thermo Fisher Scientific) and 100 μg mL^−1^ ascorbic acid. Dermal equivalents were maintained for 28 days and then harvested as appropriate for downstream processing. For the 3D culture analysis two additional housekeeping genes ACTB and GAPDH were also used for normalisation, then log fold change (LogFC) was calculated for all aged cells compared to neonatal.

### 2.5. Normalization

The raw data (containing cycle threshold values) are available for download in Appendix A. Each gene measured in each sample was normalised to peptidylprolyl isomerase A (PPIA) which is stationary throughout treatment with TGF-β. Equation (1) shows the calculation producing the 2^−ΔCT^ for each gene (g) at each timepoint. The 2^−ΔCT^ value allowed a comparison of the relative gene expression levels between samples from different conditions. To compare the magnitude of response to the TGF-β signal (the ΔΔCTg), Equation (2) further normalised the ΔCTg value by subtracting the value at time zero (CT0) from the value at time, t (CTt).
(1)$${\Delta \mathrm{CT}}_{g}=2^{-\left({\mathrm{CT}}_{g}-{\mathrm{CT}}_{\mathrm{PPIA}}\right)\;}=2^{-\Delta \mathrm{CT}}$$
(2)$${\Delta \Delta \mathrm{CT}}_{g}=2^{-\left(\left({\mathrm{CTt}}_{g}-{\mathrm{CTt}}_{\mathrm{PPIA}}\right)-\left(\mathrm{CT}0_{g}-\mathrm{CT}0_{\mathrm{PPIA}}\right)\right)\;}=2^{-\left({\mathrm{CT}}_{t}-{\mathrm{CT}}_{0}\right)\;}=2^{-\Delta \Delta \mathrm{CT}}$$

Additionally, because the adult senescent cells were not included in the initial experiment, another experiment was conducted using two of the three previous adult cell lines (H and I) plus a third cell line, J (cell line G was exhausted). To make the data comparable across experiments all data from the latter experiment were normalised by multiplication with factor γ. Factor γ was calculated as the average 2^−ΔCT^ value for experiment 1 divided by the average 2^−ΔCT^ for experiment 2, for each gene. Cell line J used the average value for each gene from cell lines H and I. However, graphs demonstrating the unnormalized 2^−ΔCT^ values for each gene in both experiments are found in Appendix A.

### 2.6. Western Blots

Fibroblasts were washed with PBS, then lysed in M-PER™ Mammalian Protein Extraction Reagent (Thermo Fisher 78501). Samples were transferred to an Eppendorf tube and were heated to 95 °C (10 min), sonicated, and centrifuged for 30 min at 10,000× *g* and the supernatant was removed. Protein (15–30 μg) was resolved on either 8%, 10%, or 12% SDS-polyacrylamide gels, dependent upon the molecular weight of the target protein. Protein was transferred onto a nitrocellulose membrane. Membranes were blocked for 45 min at room temperature with 5% non-fat milk powder in Tris-buffered saline–Tween. The following antibodies were used: collagen I (1:1000, Southern Biotech, Birmingham, AL, USA, 1310), collagen III (1:500, Proteintech, Rosemont, IL, USA, 22734-1-AP), α-SMA (1:25,000, Sigma, Kawasaki, Kanagawa, Japan, #A4416), LOXL2 (1:000, AbCam, Cambridge, UK, 179810), LTBP2 (1:250, Santa Cruz, Dallas, TX, USA, 166199), TAGLN (1:2000, AbCam, 14106), elastin (1:1000, AbCam 77804), MMP2 (1:1000, AbCam, 37150), Smad3 (1:1000, Cell Signaling, Danvers, MA, USA, 9523), P-Smad3 (1:1000, Cell Signaling, 9520), and β-actin (1:50,000, AbCam 8224), and incubated overnight at 4 °C. After washing, membranes were incubated at room temperature for 1 h with horseradish peroxidase (HRP) conjugated goat anti-mouse (1:10:000, Sigma 4416) or donkey anti-rabbit (1:10,000, Millipore, Burlington, MA, USA, AP182P) diluted in 5% milk/Tris-buffered saline–Tween. Antigens were visualized using enhanced chemiluminescence (BioRad Clarity ECL, Hercules, CA, USA, 1705061).

### 2.7. Data Analysis

All data were analysed using R version 3.6.3. Significance values shown in the tile plots were calculated by ANCOVA analysis adjusted with the Bonferroni correction. The R package, LIMMA [31] (Ritchie et al., 2015) was used to compute differential expression statistics. LIMMA was configured to use a spline matrix with 4 degrees of freedom, as per the LIMMA user guide, to compare either the control time series between adult/senescent and neonatal cell lines or the TGF-β treated cells. To make use of the available cell line replicates, differential expression calculation was repeated so that all adult/senescent cell lines were compared with all neonatal cell lines. The results were aggregated by calculating the percentage of times a gene has a false discovery rate (FDR) corrected p-value less than 0.05. This was repeated for both control and TGF-β datasets and for both adult and senescent cell lines, compared with neonatal. Because our data are dynamic, for each comparison LIMMA was configured to evaluate whether two full-time series objects were different from one other using a moderated F-statistic (Ritchie et al., 2015). Since in experiment 1, we had data from three cell lines per treatment, we made use of the available data by repeating the LIMMA analysis for all the combinations of the cell lines under a comparison. A total of 10 differential expression analyses were conducted, five each to compare TGF-β or negative control time series, two of which were ‘between’ groups comparisons and three ‘within’ groups comparisons. The output from LIMMA is a list of FDR-corrected *p*-values representing the probability that two groups are equal. Usually, a *p*-value threshold is chosen and any gene with a *p*-value lower than that threshold is considered differentially expressed. Since we had nine equivalent LIMMA analyses for the ‘between groups’ analysis and three for the ‘within groups’ analysis, a consensus strategy was used where if a gene was considered differentially expressed in >60% of LIMMA analyses, the gene was considered differentially expressed.

Using ImageJ software (version 1.53), the relative densitometry of the proteins expressed during Western analysis was calculated. Relative densitometry allows the amounts of each protein band to be observed as a ratio relative to the lane’s loading control. In brief, scanned Western films are saved as JPEG grayscale files and opened in ImageJ. With analysis measurements set to ‘grey mean only’ the rectangle tool is selected, and a box is drawn around the largest protein band; the rectangle size is set as the region of interest (ROI). Starting with the first protein lane, measurements are taken of the protein bands using the ROI. Background values are then obtained from blank areas of the film directly above or below the corresponding band. Repeat with the loading control bands and background. The pixel density is then inverted. For both the protein band and the loading control, the net value is calculated by deducting the inverted background value from the inverted band value. The final step is to take a ratio of the net band value over the net loading control of that lane. The final relative quantification values are the ratio of the net protein band of interest to the net loading control.

### 2.8. Fibroblast Migration Assay

Fibroblasts were seeded into 6 well plates at a density of 60,000 cells/cm^2^, in M106 medium (ThermoFisher Scientific, catalogue number: M106500) supplemented with LSGS (ThermoFisher Scientific, catalogue number: S00310) and incubated at 37 °C, 5% CO_2_ for 24 h. Fibroblasts were washed with PBS and serum starved for 24 h by removing LSGS from media. Cells were then treated with Mitomycin C (Sigma, Kawasaki, Kanagawa, Japan M0503) at 10 µg/mL for 2 h and washed in PBS before a scratch was made with a pipette tip and serum-free M106 was added per well. The migration of cells into the cell-free area was monitored over 48 h using a Zeiss cell observer.

## 3. Results

To identify differences between neonatal and adult fibroblasts, the activity of 72 genes were measured over 96 h in the presence or absence of TGF-β1 for three independent cell lines. Senescent cells were derived from the neonatal cell lines irradiated with 20Gy X-rays 10 days before seeding. The genes were manually selected based on a literature search to be relevant to ECM biology or TGF-β signalling. Of the 72 genes, 55 provided reliable data and were included in further analysis. 

Principal component analysis (PCA) was first conducted to gain confidence in the measurements. In the results shown in Figure 1, each point on the PCA plot represents one of the 1296 samples. The high dimensional data are reduced to fewer dimensions whilst maintaining as much of the original variance as possible. Therefore, when two points are close together on the PCA, the underlying data are similar while two distant points have more different gene expression profiles. A scree plot (Appendix A) indicates that the first three principal components account for >80% of the variance. PCA plots for the first three principal components (PCs) were coloured for cell type (neonatal, senescent, or adult, Figure 1a). Importantly, there was obvious clustering of the three cell types, with PC1 showing separation between the adult and the other two cell types, while PC2 showed separation of the neonatal from the other two cell types, and PC3 robustly separated senescent cells from neonatal and adult cell types. Cells also clustered according to cell line and showed no clustering according to biological replicate indicating no obvious batch effects (Appendix A). However, as shown in Figure 1b, samples are also clustered by timepoint. 

Differential expression analysis was conducted to investigate whether the time series measurements for each gene were statistically different between or within cell line groups. A detailed summary of the differential expression analysis is presented using a *p*-value threshold of 0.001 (Appendix A), while the number of genes that were differentially expressed in >60% of LIMMA analyses were counted and displayed in Appendix A.

Both differential expression analysis and PCA analysis indicated that aged cells showed more differences from neonatal cells than senescent cells. If ageing reflects, in large part, an increase in senescent cells and their effects [22], then this outcome is surprising. While senescence is induced in the neonatal cell lines, they are still three different cell lines, so this variable is unlikely to explain the increased similarity.

To further explore this unpredicted relationship between ageing and senescence, we analysed the data for individual genes. An additional array was conducted for the same genes using three adult cell lines and inducing senescence in these cell lines. These data were then normalised (see Section 2) to make the data comparable across the two experiments. However, un-normalised data for both experiments are shown for each gene in Appendix A. 

### 3.1. Ageing Results in Reduced Collagen Production Independent of Cell Senescence

One clear change with age was the production of collagen mRNAs *COL1A1*, *COL1A2*, *COL4A1* and *COL5A1* as shown in Figure 2a–d. Importantly, all four mRNAs showed significant decreases in adult cells when compared with both neonatal and neonatal senescent cells. There was also no significant difference in *COL4A1* between neonatal and neonatal senescent cells and a significant increase in the same mRNA from adult to adult senescent cells. All other collagen mRNAs showed a significant decrease from neonatal to neonatal senescent, but not to the same extent as in adult cells, and no significant change was observed from adult to adult senescent cells. These observations are consistent with the observations of the increased extracellular matrix (ECM) degradation associated with age [32], but inconsistent with the idea that ageing tissues lose ECM through either an increase in senescent cells [22] or progression of cells toward a “senescent-like” phenotype. 

LOXL1 and LOXL2, which are involved in collagen crosslinking [33] are also stimulated by TGF-β and showed the same significant decline in mRNA expression in adult cells compared with neonatal cells (Figure 2e,f), but *LOXL2* did not decrease with senescence while *LOXL1* significantly increased with senescence in neonatal cells. Results for LOXL2 were confirmed by Western blot (Figure 2g and Appendix A). Western blotting of collagens I and III, which are the main collagens in human skin [34,35], confirmed the production of both collagens was stimulated by TGF-β, increasing in both proliferating and senescent neonatal cells at the 96-h timepoint compared to unstimulated controls (Figure 2g,h). However, the amounts were reduced or absent in stimulated adult and adult senescent fibroblasts compared to neonatal cells. We concluded that adult fibroblasts are less fibrotic than neonatal fibroblasts both in the presence and absence of TGF-β, resulting from a mechanism independent of cell senescence.

### 3.2. Ageing and Senescence Increase Matrix Degradation

Next, we looked at the matrix metalloproteinases (MMPs) involved in collagen degradation [36]. No MMP mRNAs were significantly affected by TGF-β stimulation; however, as expected adult senescent cells showed significantly increased expression in *MMP1* (Figure 3a), *MMP2* (Figure 3b), and *MMP14* (Appendix A) compared to adult cells. What was less expected was that there was no difference between neonatal senescent and neonatal cells. Thus, the increased ECM degradation we see with age might not reflect the impact of senescent cells on ageing, but ageing on senescent cells. When we looked at the protein level (Figure 3e), MMP1 showed a similar profile to its mRNA with higher expression in adult senescent cells than any other cell types; however, MMP2 protein was highest in the neonatal senescent cells and showed some evidence of TGF-β stimulation in the neonatal and adult senescent cells. While this complicated the profile seen in the mRNA, it was not inconsistent with the idea that older tissues might undergo more matrix degradation through an increase in senescent cells, as there was an increase in MMP2 expression in adult cells compared to neonatal cells, which was not as large as the increase in the fully senescent population. 

We also looked at the expression of tissue inhibitors of metalloproteinases (TIMPs) (Figure 3c,d), which play a wide role in metalloproteinase inhibition [37]. At the mRNA level, both TIMPs showed no significant difference between aged and neonatal cells or TGF-β stimulation compared to unstimulated cells but were significantly higher in senescent cells than non-senescent cells. We confirmed this at the protein level (Figure 3e,f and Appendix A). Notably, while the increase in MMP2 in adult tissue might reflect increased senescent cells, senescent cells also produce more compensatory TIMPs, which was not true for adult cells.

Combined, these observations suggest two independent but synergistic effects of ageing fibroblasts on collagen in the ECM. Firstly, increased collagen degradation may reflect, in part, the accumulation of senescent cells, while reduced replenishment of the collagen matrix and cross-linking which protects it from degradation appear independent of cell senescence. However, these are in vitro observations from cells in monolayers. To test if ageing and senescent cells behaved similarly under more in vivo-like conditions, we constructed 3D Alvetex^®^ matrices (see Section 2) populated by cells from neonatal (<14 days), young (<30 years), middle-aged (40–45 years), and old (60+ years) individuals, as well as senescent cells (induced from neonatal cells). We then carried out an expression array of ECM genes after exposure to TGF-β (see Appendix A for all genes). The results for collagen mRNAs are shown in Figure 4a–e. *COL1A1* and *COL5A1* showed a significant decrease in expression in older cells compared to neonatal cells, while *COL1A2*, *COL3A1*, and *COL4A1* showed no significant change with age. Even for *COL1A1* and *COL5A1*, there was no evidence of continual decline in expression with age. However, all collagen mRNAs showed increased expression in senescent cells compared to neonatal cells, opposite to the effects of age. Thus, the decreases with age were unlikely to reflect higher numbers of senescent cells. The same was true for the LOXL genes (Figure 4f,g). *MMP1* showed no significant change for ageing or senescent cells (Figure 4h), while *MMP2* and *MMP14* (Figure 4i,j) showed significant increases only in senescent cells, as was the case for the TIMP genes (Figure 4k,l). These data were highly consistent with results from 2D culture, indicating that while senescent cells might be responsible for increased matrix degradation with age, reduced replacement of collagen likely reflects a senescence-independent process.

### 3.3. Ageing Cells form Fewer Myofibroblasts

We next asked why adult cells might have reduced collagen expression if not resulting from an increased number of senescent cells. We observed that the collagen and LOXL mRNAs which decreased in ageing (but not senescence) were stimulated by TGF-β, whereas the MMP and TIMP genes which were mainly affected by senescence (but not ageing) showed little effect from TGF-β stimulation. We, therefore, wondered if ageing resulted in some defect in TGF-β signaling that could have a downstream impact on ECM production. 

In canonical signaling, TGF-β binds a TGF-β receptor (TGFBR1/2) which triggers Smad2/3 phosphorylation and activation. Smad4 is recruited to the activated Smad2/3 complex and the Smad complex translocates to the nucleus where it binds to various co-factors and transcription factors to induce target gene expression [38]. One such target gene is Smad7 [39] which acts to inhibit canonical TGF-β signalling through the dephosphorylation of Smad2/3 [40], as summarized in Figure 5.

Interestingly, *TGFBR1* expression showed TGF-β-stimulated increases in all groups but adult cells (Figure 6a), suggesting that adult cells lack the positive feedback that allows them to better respond to a TGF-β signal. This was unlikely to reflect the amount of senescent cells as both *TGFBR1* and *TGFBR2* (Appendix A) expression showed a significant increase in senescent cells. Importantly, the muted TGF-β response in adult cells is unlikely to reflect a reduction in positive feedback from their reduced levels of TGFBR as the upregulation of *TGFBR1* in all cell lines is a late timepoint event (24 h+), and is thus likely a consequence of reduced TGF-β signalling rather than a cause of it. 

Similarly, the negative feedback in adult cells is also muted. *Smad3* mRNA showed a clear (though not-significant) reduction in expression in all cells stimulated by TGF-β compared to unstimulated controls. This decline usually began around 4–8 h after stimulation and likely corresponded to the increase in inhibitory *Smad7* expression which peaked initially between 2 and 3 h in response to TGF-β. However, the *Smad7* response was significantly weaker in adult cells compared to all other groups. Together, these results suggest a significantly weaker response in adult cells that does not result from cell senescence.

To investigate the underlying mechanisms further, we normalised the ΔCT values used hitherto by further subtracting the baseline value at timepoint zero for each cell line. The 2^−ΔCT^ values reflected the expression values relative to a reference gene, so they gave no indication of expression relative to the initial value. Thus, instead of 2^−ΔCT^, we used 2^−ΔΔCT^ values indicating the level of change from basal expression for each cell line. Intriguingly, the differences between neonatal and adult cells not only vanished for *COL1A1* and *COL1A2* (and were greatly reduced for *COL5A1*), but adult cells showed a trend toward higher increases in mRNAs than neonatal cells (Figure 6b and Appendix A). 

This result was unexpected as it suggested that the adult cells could increase collagen production by the same proportion (or more) as neonatal cells. Importantly, *CTGF*, a gene involved in myofibroblast formation [41,42] is significantly reduced in adult cells compared to neonatal cells (as measured by 2^−ΔCT^, Figure 6c), but the increase in proportion to basal levels (2^−ΔΔCT^) is significantly larger in adult cells compared to neonatal cells (Figure 6d). While these cells may, therefore, be perfectly functional in their capacity to respond to TGF-β, they have a reduced basal activity which means even a proportionately larger response still results in a subsequently lower level of total mRNA compared to neonatal cells. This may have important consequences. The most frequent marker of myofibroblast formation, α-SMA, encoded by the *ACTA2* gene [43,44] is significantly lower in adult cells compared to neonatal cells (2^−ΔCT^, Figure 6e), but unlike *CTGF* the proportionate increase over time is also significantly lower (2^−ΔΔCT^, Figure 6f). Importantly, Western blotting confirmed that differences in mRNA translated into differences in protein levels, including another myofibroblast marker, transgelin (TAGLN) [45,46] (Figure 6g), suggesting that adult cells form fewer myofibroblasts. 

Similar to prior observations, the reduced expression of α-SMA in adult cells is unlikely to reflect an increase in senescent cells, as neonatal senescent cells still have robust α-SMA expression at the mRNA and protein levels in response to TGF-β (Figure 6e,g). To test for functional differences, we used a migration assay, which showed a decline in the level of migration of adult cell lines compared to neonatal cell lines (Figure 6h).

We considered one final possible way that senescent cells could be responsible for the age-related reduction in collagen production and myofibroblast differentiation we see here. We and others have demonstrated a senescent cell bystander effect [47,48] by which the associated secretory phenotype (SASP) and juxtacrine signalling [49] provoke changes in the surrounding tissue. The reason cultures of purely senescent cells would not show the same decline in TGF-β response may, therefore, reflect that there are no remaining non-senescent cells to undergo the bystander effect. 

We, therefore, carried out one final experiment where neonatal and adult cells were cultured in the medium of other cell types to elucidate what effects this had on the formation of myofibroblasts. The protocol for this co-culture is described in the methods but involves the extended culture of neonatal and adult fibroblasts in the presence of medium from proliferative and senescent forms of neonatal and adult fibroblasts. As indicated by the expression of α-SMA in Western blots for neonatal cells grown in medium from other cell types (Figure 7), the capacity of neonatal cells to form myofibroblasts was not reduced by the presence of medium from proliferative adult cells or senescent neonatal or adult cells. The same profile was true for collagen I (Figure 7). Thus, we concluded that the SASP and its paracrine bystander effect are not responsible for the diminished fibrotic response of adult cells to TGF-β. However, proliferative adult cells showed reduced α-SMA expression when grown in a medium from either adult senescent or neonatal senescent cells compared to adult cells grown in either their own medium or that of proliferative neonatal cells. Thus, adult cells may have a susceptibility or altered reaction to the SASP that is not present in neonatal cells.

## 4. Discussion

Here, we demonstrate that changes in aged cells (55+-year-old donors) for collagen mRNAs and proteins are dramatically reduced compared to neonatal cells. Senescent cells from the neonatal cell lines also showed a reduction in collagen production, but not to the level of adult cells, suggesting that senescence could not be responsible for the greater decrease seen in adult cells. 

It is an increasingly common view that ageing reflects the accumulation of senescent cells and the resultant inflammation [24,25,26], and ageing cells are merely cells on the pathway toward senescence, as discussed by Ogrodnik [50]. Wlaschek et al. [22] summarised that “an emerging hypothesis of skin aging postulates that mainly fibroblast senescence drives skin decline and skin aging due to irreversible proliferation arrest and enhanced release of a senescence-associated secretory phenotype (SASP). SASP, through chemokines and proinflammatory factors, induces chronic inflammation, reduces proliferation by impaired release of essential GFs (growth factors), and enhances the degradation of the ECM (connective tissue) by enhanced activation of proteolytic enzymes, including matrix-degrading metalloproteinases”. Our data are consistent with senescent cells increasing matrix degradation, but increased degradation only threatens homeostasis if it is not accompanied by increased deposition. Instead, our data suggest that two senescence-independent processes appear to reduce the anabolic activity of old cells in response to TGF-β, at least compared to neonatal cells. Firstly, our results show that adult cells have an equal response to TGF-β as neonatal cells in terms of the proportionate increase in collagen mRNA and protein produced. However, both the baseline and stimulated production of collagens are lower. Exactly why this is, requires further investigation, but it is not because more adult cells are senescent, as senescent cells have a TGF- β response similar to neonatal cells. 

Secondly, adult cells show a markedly reduced myofibroblast capacity. Plausibly, the lower α-SMA in adult cells could reflect that the formation of myofibroblasts is a threshold event. Unlike CTGF and collagen production which rise by equal proportion in both neonatal and adult cells in response to TGF-β stimulus, because the final amount of CTGF is lower in adult cells, it may be less likely to reach the threshold required to stimulate α-SMA and induce myofibroblast differentiation, as we summarize in Figure 5. As myofibroblasts are highly important in wound healing and maintenance of the ECM, this difference may result in reduced wound healing capacity of adult cells. As senescent cells play an anti-fibrotic effect in wound healing it may exacerbate the anti-fibrotic effects of fewer myofibroblasts, but the evidence here suggests that slower wound healing with age may not be entirely related to the increase in senescent cells. 

Our co-culture results further suggest that the detrimental effect of senescent cells on tissue homeostasis may not be the result of the senescent cells themselves, but the altered response of aged cells to the SASP. Consistently, the neonatal cells show increased α-SMA and collagen I in response to adult and senescent medium, suggesting that the SASP may promote the fibrotic activity of neonatal cells and inhibit it in older cells. These results confirm the main conclusion of this paper that there is more to ageing than the simple accumulation of senescent cells and their pro-ageing effects. Adult cells behave differently to neonatal cells firstly in a way that is independent of cell senescence, and secondly in their response to senescent cells.

However, there are limitations to the conclusions drawn here. We demonstrated that adult fibroblasts are not as migratory as neonatal fibroblasts, but this was not affected by TGF-β and thus may reflect different underlying processes. Despite our attempts to replicate a more realistic skin environment using 3D culture, all conditions tested were in vitro, which has reduced applicability to human skin that results from the artificiality of the processes of isolation and in vitro growth [51,52]. That the differences we observe between ageing and neonatal cells reflect, in part, an artifact of time spent in culture cannot be ruled out, nor can we state that aged and senescent cells would necessarily behave similarly in vivo. The relevance of this work to the ageing of human skin will directly reflect how well the processes shown here translate to in vivo tissues. 

Although we attempted to confirm our mRNA results with protein levels, neither of these values are complete reflections of the activity of the molecules in question, which are also influenced by post-translational modifications and the presence and absence of inhibitors and co-factors. The amount of ECM produced and degraded cannot be accurately surmised from the results shown here, and further work is required to examine whether the changes we show would translate to the detrimental ones known to occur as a result of ageing. 

We also did not control for sex differences in the fibroblast cell lines, or timepoint after senescence induction. Senescence is not a static phenotype, changing significantly over time and according to stimulus. Very few studies address this limitation, as we discuss in a meta-analysis of the transcriptomic changes occurring after senescence induction [53]. However, the SASP is well described at the 10-day timepoint chosen here and still includes the main inflammatory factors associated with it. One of the strengths of this manuscript is that it follows the cells over an extended timescale following treatment with TGF-β, providing a detailed indication of the response that includes early and late timepoint effectors. However, it should be noted that the methods of analysis measure only the average level of expression without indicating the distribution. If α-SMA is a threshold effect as we predict, then likely some cells express high and others low levels at any given time. Lastly, because the young cells are neonatal, it is plausible that the differences reflect developmental processes rather than ageing per se. As myofibroblast formation and the TGF- β response are highly associated with wound healing, it may be that the differences we observe are the result of the rapid wound-healing response observed in neonates. However, the main differences in wound healing occur between fetal and postnatal states, with fetuses capable of rapid, scarless wound healing that is no longer possible after birth [54,55]. Although developmental differences cannot be ruled out, the distinction between ageing and development as regulators of wound healing requires further elucidation. Exactly, how ageing and development relate to each other is a key question for biogerontology, with some theories suggesting that ageing is merely the detrimental continuation of developmental processes [56]. It is certainly true that we cannot simply refer to any changes occurring before a certain point as development, and anything after as ageing.

Ageing is traditionally thought to result from accumulating molecular damage, but at least the in vitro response to TGF-β is inconsistent with this. Adult fibroblasts showed a non-significant increase in their proportionate response to TGF- β compared to neonatal cells, which would suggest that their capacity to respond was not impeded by damage. Such an observation is more consistent with a homeostatic shift toward reduced TGF-β signaling in adult cells.

One possibility is that homeostasis has shifted toward reduced ECM production and slower metabolism; consistent with a different model of ageing we have recently described as selective destruction theory (SDT) [57]. SDT predicts that many cells will undergo a reduction in metabolism that may manifest in lower basal rates of anabolic activity such as ECM production. However, whether this should result in reduced levels after stimulation will depend on the protein network. Plausibly, systems with threshold effects, as we propose for myofibroblast formation, will be particularly susceptible to failure as metabolism slows and levels of the stimulatory factors drop. However, it is also possible that thresholds could be reduced along with metabolic rate if the inhibitory factors are regulated by the same processes. Further modelling is required to address this. 

What is clear from these data is that the reduced fibrotic nature of adult cells does not directly reflect an increase in senescent cells or the effects of their SASP, so another process, be it independent damage accumulation, selective destruction, or a developmental program must be occurring.

Further work should address whether similar changes in homeostasis continue after development ceases; however, the reduction in collagen levels and increase in MMP levels are well-established markers of ageing [8,32,58]. These data are, therefore, good evidence that while senescent cells may play a role in tissue ageing and ECM degradation, cells are also undergoing an ageing process that is independent of their replicative lifespan.

## 5. Conclusions

Here, we show that senescent cells have an increased expression of MMPs, which may result in more catabolic activity and ECM degradation as we age. Conversely, the reduced anabolic activity of adult fibroblasts such as collagen production and myofibroblast formation when compared to neonatal fibroblasts, reflects a process independent of cell senescence and the SASP.

## Figures and Tables

**Figure 1 cells-13-00659-f001:**
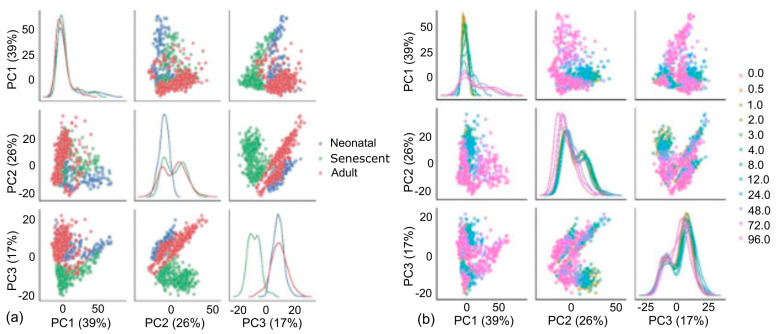
Principle component analysis of gene expression data. Each plot point reflects a summary of all gene expression for a single sample plotted for principle components 1–3 (PC1−3) which combined explain 82% of the variance. Scatter plots indicate the contribution to the variance across each PC. Kernel density plots (along the diagonal) represent the distribution of the relevant principle component. (**a**) Data are coloured by cell type. (**b**) Data are coloured by timepoint (0 h to 96 h).

**Figure 2 cells-13-00659-f002:**
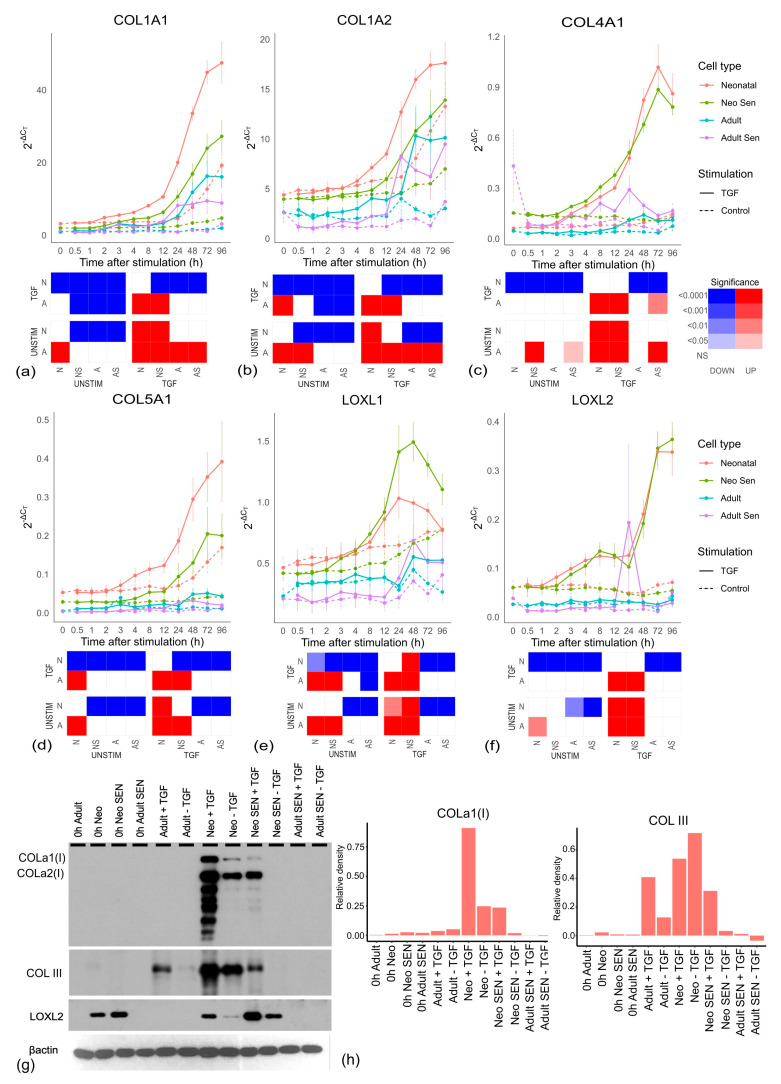
Collagen production and crosslinking by cell type. (**a**−**f**) Collagen and LOXL mRNAs for each cell line over the 96-h timecourse. Neonatal cells include non-senescent (Neonatal) and senescent (Neo Sen) cells. Adult cells include non-senescent (Adult) and senescent (Adult Sen) cells. All groups are the mean of three cell lines, which are the same lines for senescent and non-senescent cells. Each graph has an accompanying tile plot indicating whether comparison of the changing profile over time was significantly different between cell types as determined by ANCOVA. The cell types: Adult, A; Adult senescent, AS; Neonatal, N and neonatal senescent, NS were split by either TGF-β stimulation (TGF) or control (UNSTIM). Colour red indicates that the cell type along the x-axis showed a significant increase compared to the cell type and condition depicted on the y-axis, while blue reflects a significant decrease along the x-axis, and white reflects no significant change (NS). Increasing colour intensity reflects increasing significance. (**g**) Western blot of collagen I and III and LOXL2 showing protein levels at zero hours and 96 h with stimulation (+TGF) and without stimulation (−TGF) with TGF-β. (**h**) Densitometry for blots of collagen I and III (see Appendix A for LOXL2).

**Figure 3 cells-13-00659-f003:**
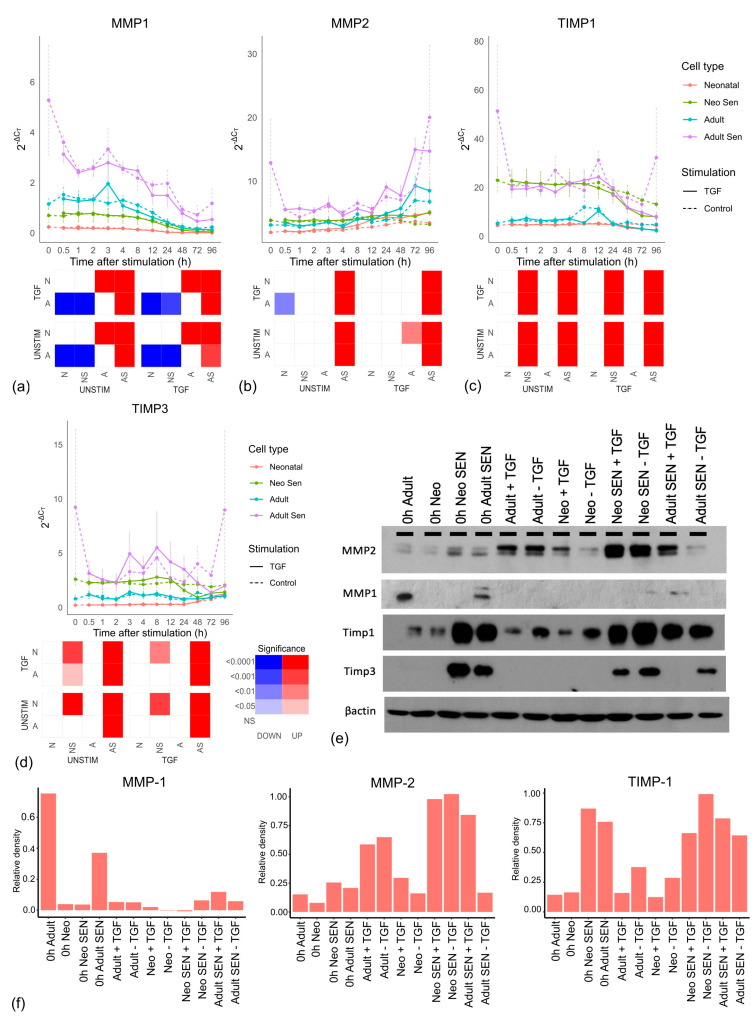
MMP and TIMP production by cell type. (**a**−**d**) MMP and TIMP mRNAs for each cell line over the 96-h timecourse. Neonatal cells include non-senescent (Neonatal) and senescent (Neo Sen) cells. Adult cells include non-senescent (Adult) and senescent (Adult Sen) cells. All groups are the mean of three cell lines, which are the same lines for senescent and non-senescent cells. Each graph has an accompanying tile plot indicating whether comparison of the changing profile over time was significantly different between cell types as determined by ANCOVA. The cell types: Adult, A; Adult Senescent, AS; Neonatal, N and Neonatal Senescent, NS were split by either TGF-β stimulation (TGF) or control (UNSTIM). Colour red indicates that the cell type along the x-axis showed a significant increase compared to the cell type and condition depicted on the y-axis, while blue reflects a significant decrease along the x-axis, and white reflects no significant change (NS). Increasing colour intensity reflects increasing significance. (**e**) Western blot of MMP1 and 2 and TIMPs 1 and 3 showing protein levels at zero hours and 96 h with stimulation (+TGF) and without stimulation (−TGF) with TGF-β. (**f**) Densitometry of western blots for MMP-2, MMP-1, and TIMP-1 (see Appendix A for TIMP-3).

**Figure 4 cells-13-00659-f004:**
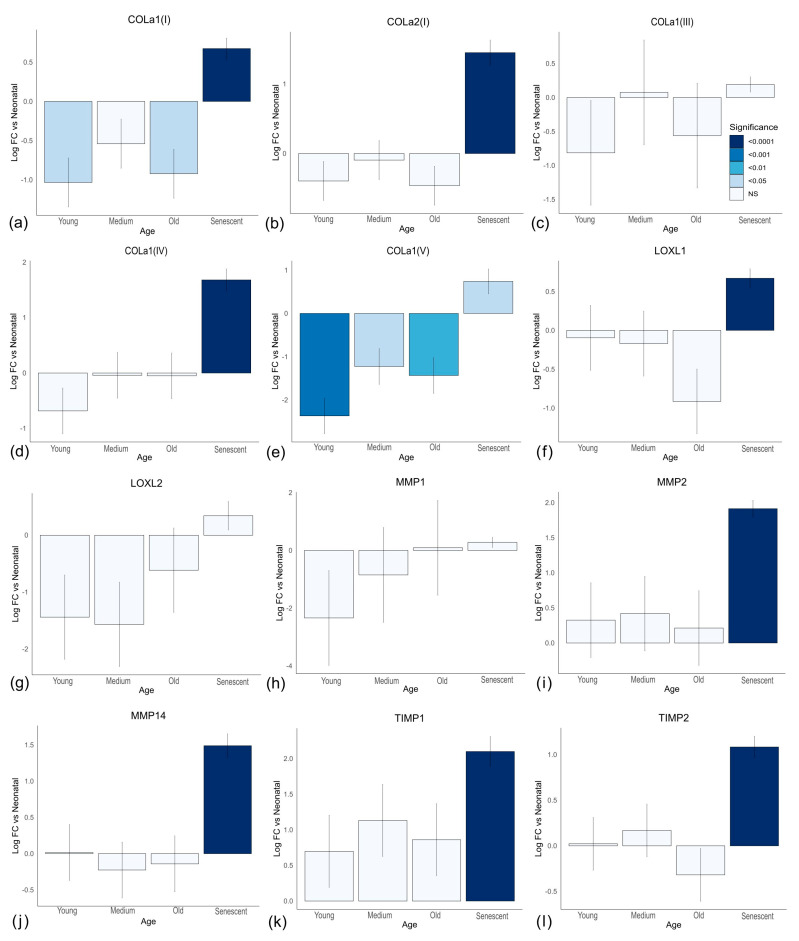
Relative gene expression for cells on 3D Alvetex^®^ matrix from different age donors and senescence. (**a**−**l**) Log fold change (Log FC) of gene expression from neonatal reflects the ΔΔCT. Each group reflects the mean of three cell lines. Neonatal cells were from donors <14 days; young cells from donors <30 years; “medium” reflects donors 40–45 years, and “old” donors were 60+ years. Senescence was induced before seeding on 3D matrix in the neonatal cell lines.

**Figure 5 cells-13-00659-f005:**
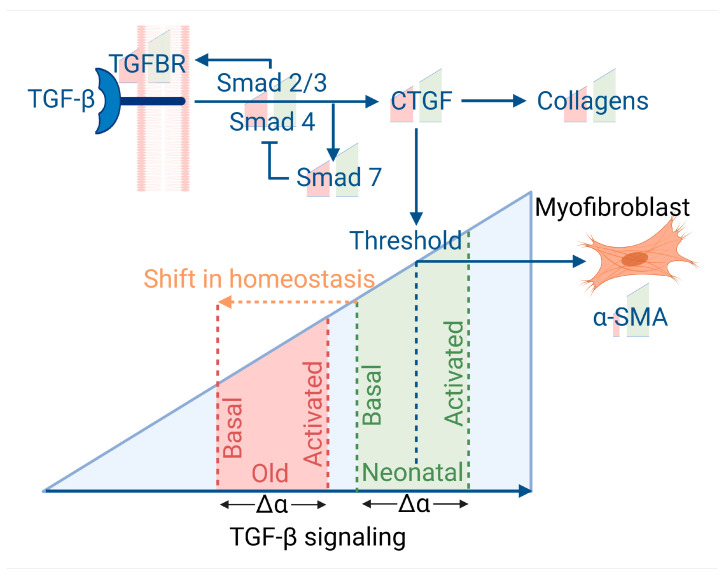
Summary of how TGF-β signaling changes with age. The use of Δα indicates that both neonatal and old cells produce a proportional increase in response to TGF-β, but the old cells have a much lower basal value, which may affect the capacity of old cells to reach the threshold required to activate myofibroblast trans-differentiation and α-SMA production. Image created with BioRender.com.

**Figure 6 cells-13-00659-f006:**
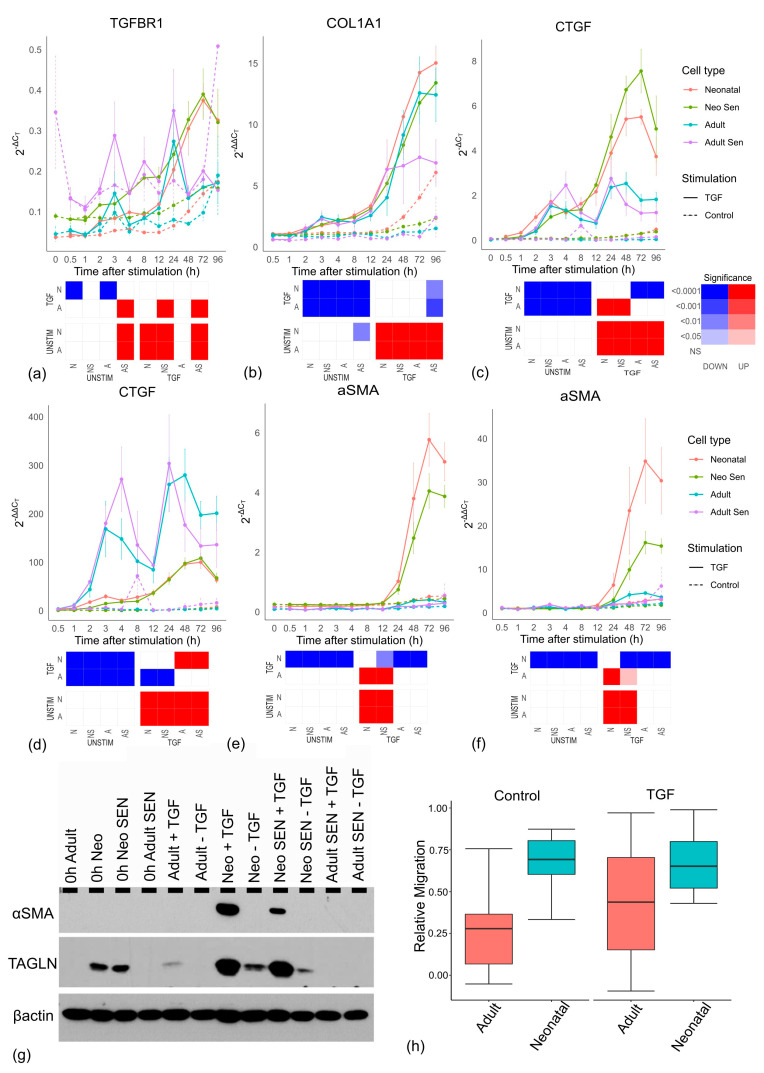
Myofibroblastic activity by cell type. (**a**−**g**) Relative mRNA levels for each cell line over the 96-h timecourse. (**a**,**d**,**f**) 2^−ΔCT^ values comparing gene expression levels between samples. (**b**,**c**,**e**,**g**) 2^−ΔΔCT^ values comparing level of change of expression from baseline at zero hours for each cell line. Neonatal cells include non-senescent (Neonatal) and senescent (Neo Sen) cells. Adult cells include non-senescent (Adult) and senescent (Adult Sen) cells. All groups are the mean of three cell lines, which are the same for senescent and non-senescent cells. Each graph has an accompanying tile plot indicating whether comparison of the changing profile over time was significantly different between cell types as determined by ANCOVA. The cell types: Adult, A; Adult Senescent, AS; Neonatal, N and Neonatal Senescent, NS were split by either TGF-β stimulation (TGF) or control (UNSTIM). Colour red indicates that the cell type along the x-axis showed a significant increase compared to the cell type and condition depicted on the y-axis, while blue reflects a significant decrease along the x-axis, and white reflects no significant change (NS). Increasing colour intensity reflects increasing significance. (**g**) Western blot of myofibroblast markers α-SMA and TAGLN showing protein levels at zero hours and 96 h with stimulation (+TGF) and without stimulation (−TGF) with TGF-β (see Appendix A for densitometry). (**h**) Relative migration over 48 h for neonatal and adult cells stimulated with TGF-β or unstimulated control.

**Figure 7 cells-13-00659-f007:**
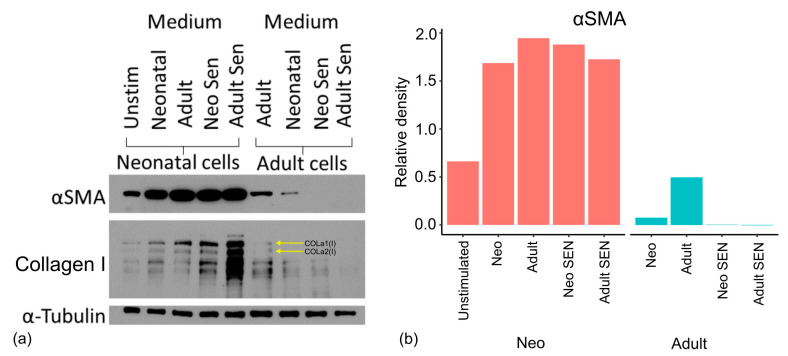
Paracrine effects on fibrotic capacity of neonatal and adult cells. (**a**) Western blots for α-SMA, collagen I, and α-tubulin. Levels of protein are for neonatal cells (left) and adult cells (right). As a control we included neonatal cells in unstimulated medium (left lane). Remaining lanes show cells stimulated with TGF-β grown in the medium of (left to right) the same cells (proliferative neonatal or adult cells), the opposite type of proliferative cells (adult or neonatal), then neonatal senescent cells and lastly adult senescent cells. (**b**) Densitometry for blots. Data are n = 1. Adult Sen, adult senescent cells; Neo Sen, neonatal senescent cells; Unstim, unstimulated cells.

## Data Availability

All data are available upon request.

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
