# Peer review of "Cell Senescence-Independent Changes of Human Skin Fibroblasts with Age"

_cells, 2024, doi:10.3390/cells13080659_

Round 1

Reviewer 1 Report

Comments and Suggestions for Authors

The research article by Fullard et al., provides an interesting expansion on the relationship between cellular senescence, cellular aging and skin aging process. The manuscript contains data from fibroblasts of several different origins, several interventions and assays and provides some novelty to the field.

Major suggestions:

1. The title, parts of the abstract and text referring to “Ageing of Human Skin” as reflecting the relevance of the provided data are misleading. While the study does use cells of dermal origin and even 3D constructs bearing certain similarities to skin, there are very remote from phenotype, function and structure of skin. It is advised to either modify the title and exercise caution when emphasizing the significance of the results for human skin aging or supplement the findings through methods like single-cell RNA sequencing (using publicly-available datasets) and histology done on human skin. Also, while I do agree that ageing of skin might be senescence-independent, stating that primarily based on the lack of evidence on senescent cells producing less collagen can appear overly conclusive.

2. I appreciate that the article delves into the complex issue of the relationship between cellular senescence, cellular aging, and differentiation. However, the article's conclusions regarding the processes believed to contribute to the observed aging-like changes in cultured cells are somewhat unclear. I found it challenging to follow the parts of the text on the induction of trans-differentiation to myofibroblasts and its connection to skin aging. This process is typically associated with healing, as myofibroblasts aid in wound contraction, rather than aging. Therefore, the conclusions about adult cells being less susceptible to TGB-b stimulated trans-differentiation compared to neonatal cells may reflect differences in healing rates between newborns and adults rather than the aging process itself. To enhance clarity, I recommend restructuring the conclusions of these experiments or providing a more detailed discussion.

Moreover, to describe better the relationship between these processes some additional data would be valuable. First, what is the basal difference in myofibroblast(-like) properties between the untreated cells of different donors/origin? Second, what is the % of fibroblasts that become myofibroblasts in control and treatment conditions (this could be addressed with FACS or IF)? Third,  what would be the hypothesis of why adult cells resist trans-differentiation to myofibroblasts if this characteristic is unrelated to cellular senescence? Lastly, incorporating a schematic or diagram illustrating the relationship between cellular senescence, cellular aging, differentiation, and their association with skin aging could enhance the manuscript's clarity and impact.

3. I find it challenging to accept that the distinction between neonatal fibroblasts and fibroblasts derived from aged individuals accurately reflects human aging. While this concern was raised in the discussion, it is possible that neonatal cells primarily represent developmental processes rather than key aspects of aging. Typically, I would recommend utilizing cells sourced from young individuals (approximately 20-30 years old); however, there are studies indicating that fibroblasts cultured in vitro may poorly mirror the properties they exhibit in aged skin (e.g., PMID: 29500330 or 9724752). Therefore, I propose a more thorough characterization of these cells to identify the underlying reasons for the observed differences. Some suggested parameters for characterization include: cell size and shape (potentially influencing migration), adhesion strength to the culture surface, proliferation rate, overall protein synthesis (which could impact collagen production), cell density in culture, spontaneous cell death, and senescence. While not mandatory, conducting advanced omics analyses such as metabolomics or epigenomics could also provide valuable insights. By establishing these fundamental characteristics of the cells used, the manuscript would benefit from a molecular understanding of the observed variances.

Minor suggestions:

1. A reference to Figure 2F appears to be missing. I would recommend that the authors thoroughly review the manuscript for similar errors.

2. I greatly appreciate that the authors made an effort to provide results for both mRNA and protein levels. However, the latter could be quantified and statistically analyzed.

3. The graphs are detailed, showing each cell line individually. However, for some figures, they and their fonts are so small that they are borderline readable. I would suggest either redoing the graphs to show an average of lines with dots representing each on the same graph or adjusting the font sizes and overall layout of the figures.

Author Response

Major suggestions:

The title, parts of the abstract and text referring to “Ageing of Human Skin” as reflecting the relevance of the provided data are misleading. While the study does use cells of dermal origin and even 3D constructs bearing certain similarities to skin, there are very remote from phenotype, function and structure of skin. It is advised to either modify the title and exercise caution when emphasizing the significance of the results for human skin aging or supplement the findings through methods like single-cell RNA sequencing (using publicly-available datasets) and histology done on human skin.

Response: we have changed the title to accommodate feedback from both reviewers

Also, while I do agree that ageing of skin might be senescence-independent, stating that primarily based on the lack of evidence on senescent cells producing less collagen can appear overly conclusive.

Response: We have modified our language throughout to make this clearer. See lines 567-574

  1. I appreciate that the article delves into the complex issue of the relationship between cellular senescence, cellular aging, and differentiation. However, the article's conclusions regarding the processes believed to contribute to the observed aging-like changes in cultured cells are somewhat unclear. I found it challenging to follow the parts of the text on the induction of trans-differentiation to myofibroblasts and its connection to skin aging.

Response: We have now added a figure and reworded this results section 3.3, which should hopefully clarify.

This process is typically associated with healing, as myofibroblasts aid in wound contraction, rather than aging. Therefore, the conclusions about adult cells being less susceptible to TGB-b stimulated trans-differentiation compared to neonatal cells may reflect differences in healing rates between newborns and adults rather than the aging process itself. To enhance clarity, I recommend restructuring the conclusions of these experiments or providing a more detailed discussion.

Response: We like this point and have added discussion on the subject. See lines 595-600.

Moreover, to describe better the relationship between these processes some additional data would be valuable. First, what is the basal difference in myofibroblast(-like) properties between the untreated cells of different donors/origin? Second, what is the % of fibroblasts that become myofibroblasts in control and treatment conditions (this could be addressed with FACS or IF)? Third,  what would be the hypothesis of why adult cells resist trans-differentiation to myofibroblasts if this characteristic is unrelated to cellular senescence? Lastly, incorporating a schematic or diagram illustrating the relationship between cellular senescence, cellular aging, differentiation, and their association with skin aging could enhance the manuscript's clarity and impact.

Response: We agree that further description of the distribution of myofibroblasts would be beneficial, and we have added discussion of this point, lines 591:594. However, we are unable to conduct the suggested experiments as the people who would conduct this research have moved on. We have however added a diagram that helps explain the relationship as suggested by the results as a new Figure 5, which should help clarify the results section 3.3 and the evaluation of the results in the Discussion.

  1. I find it challenging to accept that the distinction between neonatal fibroblasts and fibroblasts derived from aged individuals accurately reflects human aging. While this concern was raised in the discussion, it is possible that neonatal cells primarily represent developmental processes rather than key aspects of aging. Typically, I would recommend utilizing cells sourced from young individuals (approximately 20-30 years old); however, there are studies indicating that fibroblasts cultured in vitro may poorly mirror the properties they exhibit in aged skin (e.g., PMID: 29500330 or 9724752). Therefore, I propose a more thorough characterization of these cells to identify the underlying reasons for the observed differences. Some suggested parameters for characterization include: cell size and shape (potentially influencing migration), adhesion strength to the culture surface, proliferation rate, overall protein synthesis (which could impact collagen production), cell density in culture, spontaneous cell death, and senescence. While not mandatory, conducting advanced omics analyses such as metabolomics or epigenomics could also provide valuable insights. By establishing these fundamental characteristics of the cells used, the manuscript would benefit from a molecular understanding of the observed variances.

Response: we have added additional discussion about the relation of ageing and development. See lines 600-612. We have also discussed in more detail the limitations of in vitro work, including the suggested references. See lines 567-574. Unfortunately, while we would welcome further discussion, we are unable to do additional experimental work and characterization as the people working on this project have all moved on to different labs. We can only apologise for this and hope it will not be a barrier to publication. 

Minor suggestions:

A reference to Figure 2F appears to be missing. I would recommend that the authors thoroughly review the manuscript for similar errors.

Response: Yes, apologies. All figures have been checked.

  1. I greatly appreciate that the authors made an effort to provide results for both mRNA and protein levels. However, the latter could be quantified and statistically analyzed.

Response: we have added densitometry.

  1. The graphs are detailed, showing each cell line individually. However, for some figures, they and their fonts are so small that they are borderline readable. I would suggest either redoing the graphs to show an average of lines with dots representing each on the same graph or adjusting the font sizes and overall layout of the figures.

Response: This makes sense, and we have adjusted the graphs accordingly.  

Reviewer 2 Report

Comments and Suggestions for Authors

The MS by Fullard et al deals with senescence independent ageing of skin. For this they use neonatal and adult human dermal fibroblasts with are induced to undergeo senexcence via X-irradiation.

I have the following comments.

0. Title is not optimal considering the final section of discussion. Please rewrite, maybe a title including "neonatal and adult human fibroblasts". Please re-write.

1.  Senescence following irradiation should be quantified with beta-gal staining. How many % of cells became senescent following irradiation? Show staining result as figure.

2. On line 258 and onwards , collagens are written with roman numerals,:

-  not collagen 1 and 3 but collagen I and III,

This goes for figures  (1 and 6)

-not 1a1 but a1(I)

-not 1a2   but a2(I)

-not col3  but a1(III)

-col 1 but a1(I)

3. MMPs are enzymes, and what "counts" here  is enzymology. If MMP1 ( cleaves native collagen) and MMP2 ( activity can be determined in gelatin loaded gel) activities are not evaluated, all the results with MMPS need to be toned down. Activity of MMPs should be determined.

4. line 336 Language use: "Strangely, MMP1 showed not significant". Please re.-write, Strangely is unscientific and does not fit into this context. It might be unexpected (in that case explain why unexpected), but it is not strange...

5. The discussion in line 461- 469 should move into discussion section.

Author Response

The MS by Fullard et al deals with senescence independent ageing of skin. For this they use neonatal and adult human dermal fibroblasts with are induced to undergeo senexcence via X-irradiation.

I have the following comments.

Title is not optimal considering the final section of discussion. Please rewrite, maybe a title including "neonatal and adult human fibroblasts". Please re-write.

Response: we have changed the title to accommodate feedback from both reviewers

  1. Senescence following irradiation should be quantified with beta-gal staining. How many % of cells became senescent following irradiation? Show staining result as figure.

Response: we have added lines explaining this in the methods and referenced the papers where we characterised the senescence response in these cells to irradiation. See lines 90-92.

  1. On line 258 and onwards , collagens are written with roman numerals,:

-  not collagen 1 and 3 but collagen I and III,

This goes for figures  (1 and 6)

-not 1a1 but a1(I)

-not 1a2   but a2(I)

-not col3  but a1(III)

-col 1 but a1(I)

Response: apologies, changed throughout.

MMPs are enzymes, and what "counts" here  is enzymology. If MMP1 ( cleaves native collagen) and MMP2 ( activity can be determined in gelatin loaded gel) activities are not evaluated, all the results with MMPS need to be toned down. Activity of MMPs should be determined.

Response: agreed, we have added lines 575-581 in the limitations to discuss this point.

line 336 Language use: "Strangely, MMP1 showed not significant". Please re.-write, Strangely is unscientific and does not fit into this context. It might be unexpected (in that case explain why unexpected), but it is not strange...

Response: thanks, changed.

  1. The discussion in line 461- 469 should move into discussion section.

Response: Done

Round 2

Reviewer 1 Report

Comments and Suggestions for Authors

Authors addressed my comments sufficiently

Author Response

Thanks

Reviewer 2 Report

Comments and Suggestions for Authors

Unfortunately, the authors have not managed with the right nomenclature for collagen in neither text of figures. Roman numerals not used correctly, symbols lacking for alpha chains etc. Error in all possible combination gives a very sloppy impression throughout.

Please see table below for what corrections need to be done :

Protein chain

mRNA/gene

Text/figure with errors

collagen a1(I)

1A1

sections 3.1, figure 1f, figures 2a-c, g, figure 6

collagen a2 (I)

1A2

sections 3.1, figure 1f, figures 2a-c, g, figure 6

collagen a1 (III)

3A1

collagen a1(V)

5A1

sections 3.1, figure 1f, figures 2a-c, g,

collagen a1(IV)

4A1

Author Response

We have addressed all concerns about the collagen RNAs and proteins in Figs and text.